# Cable-Driven Unmanned Aerial Manipulator Systems for Water Sampling: Design, Modeling, and Control

Li Ding [1,*,†] , Guibing Zhu [2,†] , Yangmin Li [3] and Yaoyao Wang [4]

1   College of Mechanical Engineering, Jiangsu University of Technology, Changzhou 213001, China
2   Maritime College, Zhejiang Ocean University, Zhoushan 202450, China; zhuguibing2003@163.com
3   Department of Industrial and Systems Engineering, The Hong Kong Polytechnic University,
    Hong Kong 999077, China; yangmin.li@polyu.edu.hk
4   National Key Laboratory of Science and Technology on Helicopter Transmission,
    Nanjing University of Aeronautics and Astronautics, Nanjing 210000, China; yywang-cmee@nuaa.edu.cn
*   Correspondence: nuaadli@163.com
†   These authors contributed equally to this work.

**Abstract:** The unmanned aerial manipulator (UAM) is a kind of aerial robot that combines a quadrotor aircraft and an onboard manipulator. This paper focuses on the problems of structure design, system modeling, and motion control of an UAM applied for water sampling. A novel, light, cable-driven UAM has been designed. The drive motors installed in the base transmit the force and motion remotely through cables, which can reduce the inertia ratio of the manipulator. The Newton–Euler method and Lagrangian method are adopted to establish the quadrotor model and manipulator model, respectively. External disturbances, model uncertainty, and joint flexibility are also accounted for in the two submodels. The quadrotor and manipulator are controlled separately to ensure the overall accurate aerial operation of the UAM. Specifically, a backstepping control method is designed with the disturbance observer (BC-DOB) technique for the position loop and attitude loop control of the quadrotor. A backstepping integral fast terminal sliding mode control based on the linear extended state observer (BIFTSMC-LESO) has been developed for the manipulator to provide precise manipulation. The DOB and LESO serve as compensators to estimate the external disturbances and model uncertainty. The Lyapunov theory is used to ensure the stability of the two controllers. Three simulation cases are conducted to test the superior performance of the proposed quadrotor controller and manipulator controller. All the results show that the proposed controllers provide better performances than other traditional controllers, which can complete the task of water quality sampling well.

**Keywords:** aerial manipulator; water sampling; mechanical design; system modeling; motion control





## 1. Introduction

Water quality monitoring plays an important role in many circumstances, such as tracking changes in water quality over time, identifying specific existing or emerging water quality problems, and periodically assessing water quality [1–3]. Physical, chemical, and bacteriological analysis of water samples is crucial for water quality monitoring. Water sampling faces various challenges, including a lack of personnel, limited access to water bodies, and time constraints, particularly during natural disasters and emergencies. In addition, the quality of water samples significantly influences the analysis results. Depending on the analysis, the delivery time of water samples to a laboratory is also important. Ideally, within a few hours of collection, all water samples should be delivered to a central or regional laboratory [4]. However, this situation depends on the security of vehicles for sampling officers and the quality of the transportation system. But these services are not widely accessible in many regions and countries. To address these issues,

intelligent equipment and advanced technologies have been developed for autonomous water sampling from water bodies.

In recent years, UAMs have attracted great attention in academia and industry. They can offer aerial platforms (e.g., multirotors and helicopters) [5] equipped with a wide range of robotic manipulators capable of physically interacting with the surroundings, which has expanded the capability of active operations for unmanned aerial vehicles. To this date, UAMs can execute some tasks where human access is restricted, such as aerial operation and grasp [6], inspection and maintenance [7], collaboration with ground robots [8], transportation and position [9], and canopy sampling [10]. Motivated by this, UAMs can also be used for water sampling, especially around drain outlets, to ensure the reality of the water sample. As a complex robotic system, UAMs developed for water sampling face several significant challenges, such as structural design, system modeling, and motion control.

UAMs are complex multibody systems exhibiting coupled dynamic behavior, which should be considered in the design of their components. Kondak et al. [11] developed an aerial manipulator with a total weight of 120 kg, composed of an autonomous helicopter and a seven-degree-of-freedom (DOF) industrial manipulator. The overweight can adversely affect the mission performance regarding payload capacity, working range, and control disturbances. Jimenez-Cano et al. [12] chose a large-size helicopter as a platform to equip a heavy, multilink robotic arm. Designing an aerial manipulator system involves balancing the trade-offs between aerial mobility and manipulation capabilities, as well as considering factors such as power consumption, payload capacity, and control system stability. For the common low-price drones with weak load capacity, lightweight features play a critical role in aerial manipulator design. However, an apparent common shortcoming in the mentioned applications is that UAMs use high-weight robotic manipulators to perform tasks, but flight time is strictly shortened. The drive components of conventional unmanned aerial manipulators are mounted at the joints, resulting in high inertia and stiffness [13–15]. A cable-driven mechanism has been integrated into unmanned aerial manipulators to cope with the above problems. The mechanism offers less inertia, higher flexibility, and better safety for operating objects by rearranging drive components and utilizing flexible cables to convey motion and force. The novel kind of prototype is commonly called a cable-driven aerial manipulator. Furthermore, a UAM with a light cable-driven manipulator will be designed for water quality sampling in this paper.

The first challenge in UAM research is dynamics modeling. The modeling methods of UAMs contain integral modeling and independent modeling [16]. In integral modeling, the motion of each rigid body of the system is represented by the motion of a multilinked rigid body with a floating base, which is first studied in the field of space manipulators. For such complex dynamics modeling, Euler–Lagrange equations are mostly used, and the complete rigid body dynamics model obtained is very complex and computationally intensive. For example, Abaunza et al. developed a UAM with a 2-DOF manipulator, and derived the kinematic and dynamical equations of the whole system by combining the Newton–Euler method [17]. Tomasz et al. used the Lagrangian method to obtain the analytical solutions of the generalized forces and moments of a UAM, and obtained dynamical models [18]. The integral modeling approach ignores the changes in the center of gravity and inertia of the manipulator during operation, and directly considers the coupling terms as internal factors of the system, which can lead to a decrease in the accuracy of the modeling. The independent modeling approach treats the coupling effects between the aircraft and the manipulator as external disturbances, and models them separately [19]. The dynamics model created by this method is not as complex as the holistic modeling approach, simplifying the modeling and control process. In our work, when the cable-driven aerial manipulator is in water sampling mode, the aircraft, in hover mode, is treated as a floating platform, and only the dynamics of the manipulator are considered. In flight mode, the manipulator serves as the payload of the aircraft, and only the dynamics of the

aircraft are considered. Therefore, this paper intends to adopt an independent modeling method to obtain the system model of a UAM.

Another challenge for UAMs is controller design due to their complex dynamics. In some papers, the aircraft and manipulator are regarded as a single system for the purposes of control. A proportion integration differentiation (PID) controller was designed for a UAM to complete the grasping task [20]. A decoupled adaptive controller based on Lyapunov theory was adopted to eliminate the effect produced by the manipulator of the UAM [21]. Martin et al. proposed a variable-parameter integral inversion method to express the rotational inertia and center of mass of the aircraft as a function of the joint angle of the manipulator, and compensate for the motion of the aircraft in manipulator control [22]. In addition, there are other control algorithms, such as feedback linearization [23], the linear quadratic regulator control (LQR) algorithm [24], fuzzy control [25], nonlinear inverse control [26], model predictive control [27], and sliding mode control (SMC) [28]. Among them, SMC is widely used in the control of electromechanical systems because of its strong robustness, simple structure, and insensitivity to parameters. However, the SMC structure contains switching functions, which cause the chattering phenomenon. Therefore, Ma et al. proposed the terminal sliding mode control (terminal SMC, TSMC) by adding higher-order nonlinear functions to the sliding mode surface, which effectively weakened the chattering, but it also posed discontinuity and singularity problems [29]. Further, Yi et al. proposed a fast continuous nonsingular terminal sliding mode control strategy (fast nonsingular TSMC, FNTSMC) to solve the singularity problem and enhance the convergence of the system state [30]. In addition, integral TSMC (ITSMC) can guarantee robustness by obtaining a suitable initial position so that the system has only a sliding phase, which provides a convergence in finite time and fast transient response [31]. However, the lumped disturbances consisting of internal uncertainties and external perturbations existing in the cable-driven manipulator affect the steady-state performance of the joint variables, thus reducing the overall control quality of the system. According to the references [32,33], the state observer can effectively estimate and compensate for the lumped disturbances and improve the system's resistance to disturbances. Among the state observers, the linear extended state observer (LESO) has the characteristics of low energy consumption and easy engineering implementations, and is successfully embedded in the structures of backstepping control (BC) [34], adaptive control [35], and PD (proportion differentiation, PD) control [36]. Based on the analyses mentioned above, this paper intends to combine the advantages of ITSMC, BC, and LESO to design a motion controller for the cable-driven manipulator. Meanwhile, a disturbance observer (DOB) is introduced to estimate the disturbances of the aircraft, and the BC method is used to ensure the accuracy of the position and attitude of the aircraft.

A UAM for water sampling should be low-complexity, simple to operate, and effective from both a commercial and technical aspect. This paper focuses on the structural design, system modeling, and controller design of a UAM, all of which have research value and importance. The main contributions of this work are summarized as follows:

(i) We designed a flying robot equipped with a cable-driven aerial manipulator to collect water samples at the drain outlets. This design can effectively reduce the weight of the robotic arm and joint inertia, and improve the duty ratio of the end effector. As a result, our robotic arms are lightweight, dexterous, and capable of a fast response.

(ii) Compared with SMC schemes [37,38], a backstepping integral fast terminal sliding mode control based on the linear extended state observer (BIFTSMC-LESO) for the cable-driven manipulator is designed for the first time. The hybrid controller ensures that the state quantities can converge in finite time, and has better transient and steady-state performance.

(iii) Several practical factors, such as external disturbances, and internal unmodeled characteristics are considered in our work. We use DOB to observe the lumped disturbances for the quadrotor, and use the LESO to estimate the lumped disturbances for the

manipulator, respectively. It can ensure stable tracking without information on the system compared with other controllers [27,39].

The rest of this paper is organized as follows. Section 2 presents the mechanical design of the UAM. The system model is established in Section 3. Section 4 describes the controller design for the UAM. Section 5 covers the simulation cases and results. The conclusions and suggestions for future work are shown in Section 6.

## 2. Mechanical Design

The 3D virtual model of the developed UAM is shown in Figure 1, which contains three main components, i.e., unmanned quadrotor, water sampling system, and cable-driven manipulator. The working principle of the prototype is to control the quadrotor to hover near the drain outlets, then manipulate the cable-driven manipulator to insert its end effector into the pipe mouth to collect water samples.

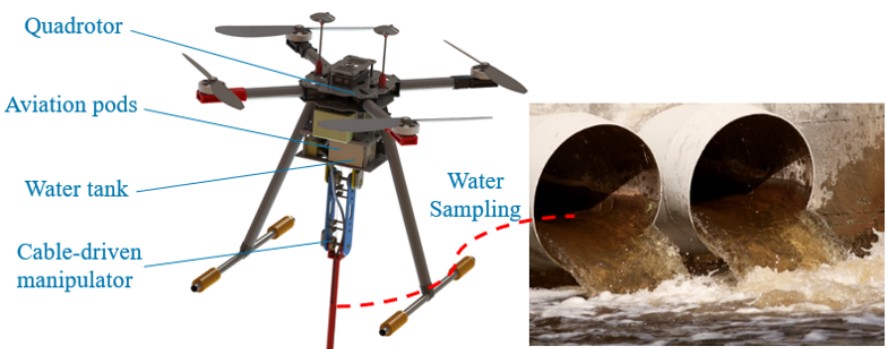

**Figure 1.** A 3D virtual model of the aerial manipulator.

The aerial platform selected is an X450 quadrotor that has robust autonomous hovering capability with a minimum drift of position, and is well suited for positioning and navigational control strategies, which can increase the operational capability of the manipulator. It is equipped with a set of avionics, such as a flight controller, two pairs of motors and propellers, four electronic speed controllers, and a global position system (GPS). The GPS provides absolute positioning with respect to world coordinates, while inertial sensors provide required data for the attitude controller. In addition, the lithium battery, water pump, and water tank are placed in the aviation pods.

As illustrated in Figure 2, the length of the fully extended robotic arm is 515 mm, the lengths of links 1 and 2 are 115 mm and 150 mm respectively, and the length of the end effector is 250 mm. The manipulator provides a light arm with cable-driven mechanisms that has two parallel joints. Each joint is driven by a DC geared motor installed in the aviation pods. A pair of driven cables (red and blue lines) are provided to control a joint rotating in two directions, which are kept under tension by the tension wheels. As a result, the joints can be controlled remotely through the driving wheels and guide wheels. Moreover, the manipulator also incorporates a suction pipe that draws water into the water tank installed in the aviation pods.

The cable-driven mechanism is described as follows by taking joint 2 as an example. As shown in Figure 3, joint 2 is rotated clockwise by the red cable and counterclockwise by the blue cable. The torque produced by DC geared motor 2 is transmitted from driving wheel 2 to joint wheel 2 through the guide wheel. Starting at driving wheel 2, the red cable goes clockwise around driving wheel 2 before wrapping counterclockwise around guiding wheel 1. Afterwards, the red cable wraps around joint wheel 2 in a counterclockwise direction after going around two tension wheels in opposite directions. This completes the winding arrangement of a driving cable. As a result, joint 2 is driven clockwise by the red cable. Similarly, joint 2 rotates counterclockwise through the blue cable.

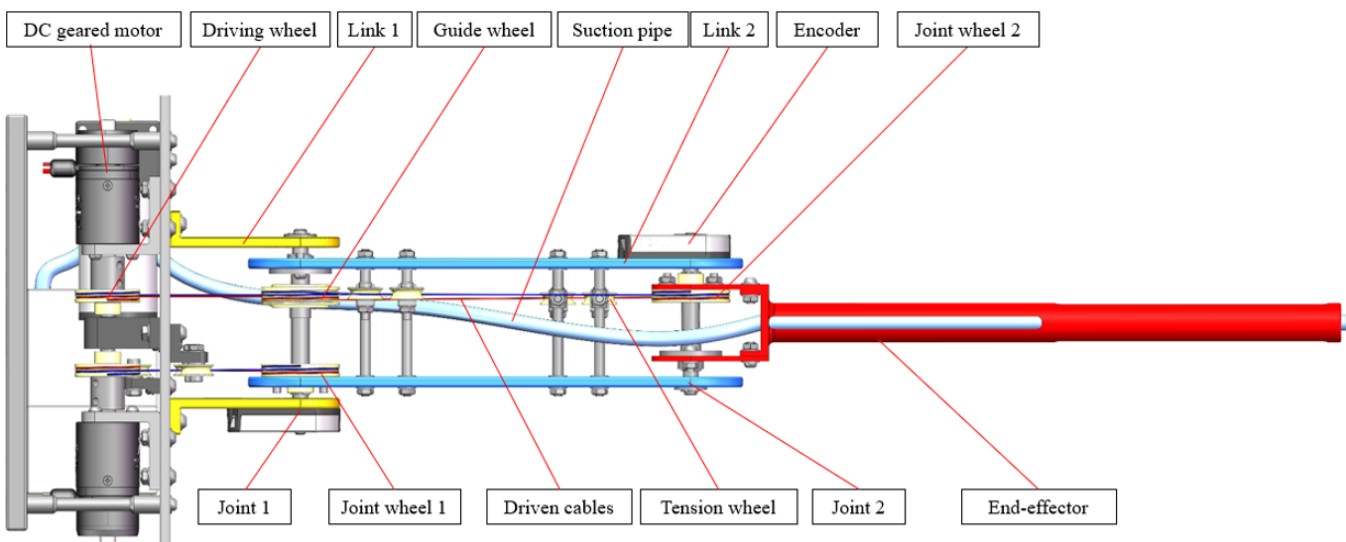

**Figure 2.** Mechanical structure of the cable-driven manipulator.

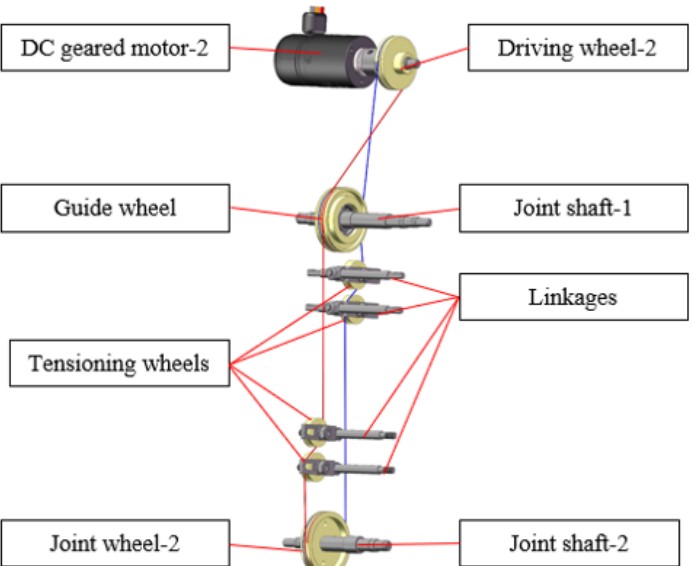

**Figure 3.** Cable-driven mechanism.

The inner structure of the aviation pods is shown in Figure 4, which reveals the water collection mechanism. The water from the drain outlets is collected through the suction pipe, and flows into the water tank through the drain pipe. A water pump provides the power to ensure the wastewater can be pumped from lower levels to higher levels. The water pump is driven by a drive motor. The size of the water tank is 100 mm, which can hold about 1 L of wastewater.

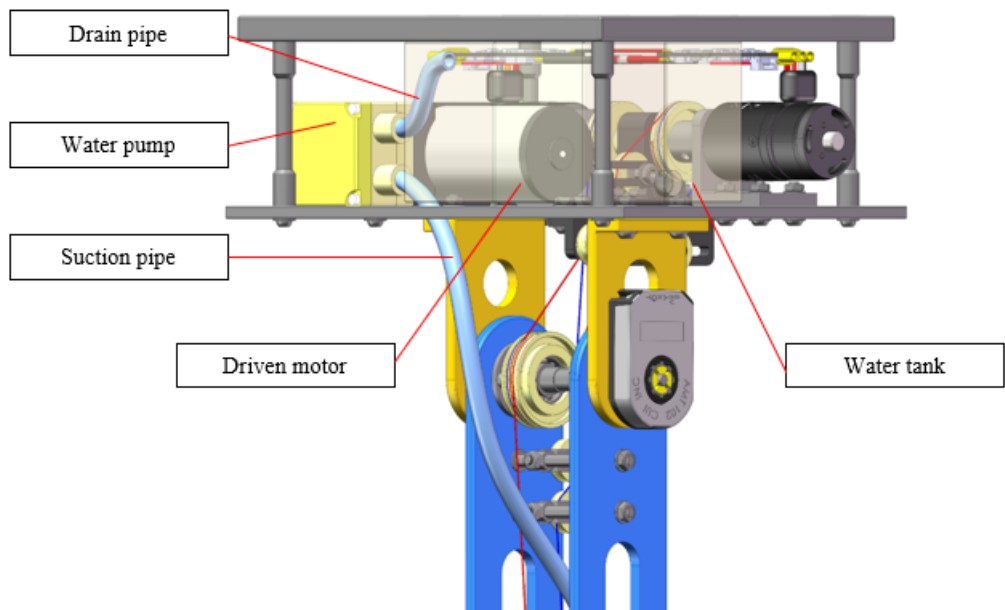

**Figure 4.** Structure of the aviation pods.

### 3. System Modeling

**Remark 1.** *The developed aerial manipulator is divided into two submodels, namely, a quadrotor model and a serial manipulator with two degrees of freedom. The coupling effect between the two submodels can be ignored during the modeling process, but treated as parametric uncertainties during controller design.*

Three coordinate frames are used to describe the system: inertial coordinate frame $\{I\}$, body coordinate frame $\{B\}$, and manipulator coordinate frames $\{1\}$, $\{2\}$, and $\{e\}$. Since the quadrotor is a rigid 6-DOF object, its dynamics can be computed by applying the Newton–Euler method. Here is the mathematical model for the quadrotor:

$$
\begin{cases}
\ddot{x} = \left[ (\sin\theta\cos\psi\cos\phi + \sin\psi\sin\phi)U_1 - k_x\dot{x}|\dot{x}| \right]/m \\
\ddot{y} = \left[ (\sin\theta\sin\psi\cos\phi - \cos\psi\sin\phi)U_1 - k_y\dot{y}|\dot{y}| \right]/m \\
\ddot{z} = \left[ \cos\theta\cos\phi U_1 - k_z\dot{z}|\dot{z}| \right]/m - g \\
\ddot{\phi} = \left[ (I_{yy} - I_{zz})\dot{\theta}\dot{\theta} + U_2 - k_\phi\dot{\phi}|\dot{\phi}| \right]/I_{xx} \\
\ddot{\theta} = \left[ (I_{zz} - I_{xx})\dot{\psi}\dot{\phi} + U_3 - k_\theta\dot{\theta}|\dot{\theta}| \right]/I_{yy} \\
\ddot{\phi} = \left[ (I_{xx} - I_{yy})\dot{\phi}\dot{\theta} + U_4 - k_\psi\dot{\psi}|\dot{\psi}| \right]/I_{zz}
\end{cases}
\tag{1}
$$

where $[x, y, z]^T$ and $[\phi, \theta, \psi]^T$ denote the position and attitude of the quadrotor, respectively. The term $[I_{xx}, I_{yy}, I_{zz}]^T$ is the inertia of the axes $x$, $y$, and $z$, respectively. The term $k_i(i = x, y, z)$ is the drag coefficient and $k_j(j = \phi, \theta, \psi)$ is the aerodynamic friction factor, $m$ is the mass of the quadrotor, and $g$ is the gravitational acceleration. $U_i(i = 1, 2, 3, 4)$ is the control input, which satisfies the below relationship with the angular speeds $\Omega_i(i = 1, 2, 3, 4)$ as follows:

$$
\begin{bmatrix} U_1 \\ U_2 \\ U_3 \\ U_4 \end{bmatrix} = \begin{bmatrix} k_t & k_t & k_t & k_t \\ \frac{\sqrt{2}}{2}k_tL & -\frac{\sqrt{2}}{2}k_tL & -\frac{\sqrt{2}}{2}k_tL & \frac{\sqrt{2}}{2}k_tL \\ \frac{\sqrt{2}}{2}k_tL & \frac{\sqrt{2}}{2}k_tL & -\frac{\sqrt{2}}{2}k_tL & -\frac{\sqrt{2}}{2}k_tL \\ -k_m & k_m & -k_m & k_m \end{bmatrix} \cdot \begin{bmatrix} \Omega_1^2 \\ \Omega_2^2 \\ \Omega_3^2 \\ \Omega_4^2 \end{bmatrix}
\tag{2}
$$

where $k_t$ and $k_m$ are the thrust coefficient and torque coefficient, respectively. $L$ is the distance between the rotation axes and the center of the quadrotor.

**Remark 2** ([40]). *In this paper, the quadrotor takes flights near the hovering state. In this case, one observes that $\theta \approx 0$, $\phi \approx 0$, $\sin \phi \approx 0$, $\sin \theta \approx 0$, $\cos \phi \approx 1$, $\cos \theta \approx 1$. The yaw angle is not controlled frequently, so $\dot{\psi} \approx 0$ can be obtained. Meanwhile, since the rotary inertia is small and the quadrotor is symmetric, one observes that $I_{xx} \approx I_{yy}$. It should be noted that the linear model can describe a small range of flight modes, including hovering, low-speed flight, takeoff, and landing. Although there are some limitations, it can be used to describe the motion of the proposed aerial manipulator in this paper.*

Under Remark 2, the dynamic model (1) can be simplified to the following form:

$$\begin{cases} \ddot{x} = \frac{U_1}{m}(\theta \cos \psi + \phi \sin \psi) - \frac{k_x \dot{x}|\dot{x}|}{m} \\ \ddot{y} = \frac{U_1}{m}(\theta \sin \psi - \phi \cos \psi) - \frac{k_y \dot{y}|\dot{y}|}{m} \\ \ddot{z} = \frac{U_1}{m} - g - \frac{k_z \dot{z}|\dot{z}|}{m} \\ \ddot{\phi} = \frac{U_2 - k_\phi \dot{\phi}|\dot{\phi}|}{I_{xx}} \\ \ddot{\theta} = \frac{U_3 - k_\theta \dot{\theta}|\dot{\theta}|}{I_{yy}} \\ \ddot{\psi} = \frac{U_4 - k_\psi \dot{\psi}|\dot{\psi}|}{I_{zz}} \end{cases} \tag{3}$$

**Assumption 1.** *For a cable-driven aerial manipulator, the motor transmits power to the joint along flexible cables so that the effect of the flexible cables can be equated with flexible joints. The flexibility of the joint is provided by a linear torsional spring system. Joint force and moment can be regarded as linearly related to joint flexibility variation.*

**Assumption 2.** *The joint flexibility also contains hysteresis, joint clearance, and other nonlinear factors.*

**Assumption 3.** *The motor rotors can be considered uniform cylinders.*

With Assumptions 1–3, the dynamics model of the cable-driven aerial manipulator considering joint flexibility in non-contact mode is described as

$$M(q)\ddot{q} + C(q, \dot{q})\dot{q} + G(q) + I_m \ddot{\theta} + D_m \dot{\theta} = \tau + \tau_d \tag{4}$$

where $\theta$, $\dot{\theta}$, $\ddot{\theta}$, $I_m$, $D_m$, and $\tau$ are the position, velocity, acceleration, inertia, damp, and input torque of the motors, respectively. $q$, $\dot{q}$, and $\ddot{q}$ are the position, velocity, and acceleration of the joints, respectively. $M$, $C$, and $G$ are the inertia matrix, centrifugal and Coriolis forces term, and gravity term, respectively. $\tau_d$ is the external disturbance.

Furthermore, we add the uncertain terms $M_0$, $C_0$, and $G_0$ into Equation (4), and the dynamics model can be rewritten as

$$\tau = M(q)\ddot{q} + F + \tau_d \tag{5}$$

where $F = C(q, \dot{q})\dot{q} + G(q) + I_m \ddot{\theta} + D_m \dot{\theta} + M_0 \ddot{q} + C_0 \dot{q} + G_0$ is the system function that contains the internal unmodeled characteristics. $f = F + \tau_d$. $F$ is the lumped disturbances.

## 4. Controller Design

This section is divided into subheadings. It provides a concise and precise description of the experimental results, their interpretation, as well as the experimental conclusions that can be drawn.

### 4.1. Quadrotor Controller Design

Based on the quadrotor dynamics [41], a dual-loop controller is designed to achieve its trajectory tracking control. Position loops track the quadrotor's 3D trajectory, and attitude loops stabilize its attitude. When the quadrotor arrives at the water quality sampling point, it is necessary to keep the position of the quadrotor stable in order to ensure the quality of the operation.

According to references [42,43], the position dynamics of the quadrotor in Equation (3) can be described as

$$\ddot{\boldsymbol{P}} = P1_0(\boldsymbol{x}) + P2_0(\boldsymbol{x})\boldsymbol{u}_P \tag{6}$$

where the nominal expressions of $P1_0(\boldsymbol{x})$ and $P2_0(\boldsymbol{x})$ are given by

$$P1_0(\boldsymbol{x}) = \frac{-1}{m} \begin{bmatrix} k_x \dot{x}|\dot{x}| & k_y \dot{y}|\dot{y}| & k_z \dot{z}|\dot{z}| \end{bmatrix}^T \tag{7}$$

$$P2_0(\boldsymbol{x}) = \frac{1}{m} \begin{bmatrix} U_1 \cos\psi & U_1 \sin\psi & 0 \\ U_1 \sin\psi & -U_1 \cos\psi & 0 \\ 0 & 0 & \cos\phi\cos\theta \end{bmatrix} \tag{8}$$

and the control signals of the position are defined as

$$\boldsymbol{u}_P = \begin{bmatrix} \theta, & \phi, & U_1 \end{bmatrix}^T \tag{9}$$

Firstly, the position tracking error and velocity tracking error are defined as

$$z_{P,1} = \boldsymbol{P} - \boldsymbol{P}_r \tag{10}$$

$$z_{P,2} = \dot{\boldsymbol{P}} - \boldsymbol{\alpha}_1 \tag{11}$$

where $\boldsymbol{P}$ is the measured position, $\boldsymbol{P}_r$ denotes the referenced position, $\boldsymbol{\alpha}_1$ is the virtual control signal.

The derivative of Equation (11) is defined as

$$\dot{z}_{P,1} = \boldsymbol{\alpha}_1 + z_{P,2} - \dot{\boldsymbol{P}}_r \tag{12}$$

Based on the common PI control law, the virtual control signal $\boldsymbol{\alpha}_1$ is defined as

$$\boldsymbol{\alpha}_1 = -k_{p,1}\dot{z}_{P,1} - k_{p,2}z_{P,1} + \ddot{\boldsymbol{P}}_r = -k_{p,1}(\boldsymbol{P} - \boldsymbol{P}_r) - k_{p,2}z_{P,1} + \ddot{\boldsymbol{P}}_r \tag{13}$$

where the control parameters $k_{p,1}$ and $k_{p,2}$ are positive numbers.

Then, the derivative of Equation (13) yields

$$\dot{\boldsymbol{\alpha}}_1 = -k_{p,1}\dot{\sigma}_{P,1} - k_{p,2}z_{P,1} + \dot{\boldsymbol{P}}_r \tag{14}$$

According to Equations (10) and (11), the linear velocity of the quadrotor is defined as

$$\dot{z}_{P,2} = \zeta_0(\boldsymbol{x}) + \zeta_0(\boldsymbol{x})\boldsymbol{u}_P - \dot{\boldsymbol{\alpha}}_1 + d(\boldsymbol{x}, t) \tag{15}$$

where $d(\boldsymbol{x}, t)$ is the disturbance of the aircraft system.

The input compensation is defined as $\hat{d}(\boldsymbol{x}, t) = d(\boldsymbol{x}, t)/P2_0(\boldsymbol{x})$ to resist the external disturbances. When there is non-continuous and high-frequency noise in $1/P2_0(\boldsymbol{x})$, a low-frequency filter called a Q-filter can be used [42]. In our work, since the UAM is in hover mode or low-speed flight, $/P2_0(\boldsymbol{x})$ is a nonsingular constant matrix, so the filter design only needs to consider the filtering of noise. As the position control loop is coupled to the attitude control loop, a filter in the form of an integral can be used when the attitude control loop is not accurately identified.

$$\hat{d}(\boldsymbol{x}, t) = \frac{z_{P,2} + \boldsymbol{\alpha}_1 - \int_0^t (P1_0(\boldsymbol{x}) + P2_0(\boldsymbol{x})\boldsymbol{u}_P)dt}{t} \tag{16}$$

Therefore, the position tracking controller can be defined as

$$u = \frac{-k_{p,3}z_{P,2} + \dot{\alpha}_1 - \zeta_0(x) - \hat{d}(x,t)}{\varsigma_0(x)} \tag{17}$$

where $k_{p,3} > 0$ is the control parameter. $\hat{d}(x,t)$ is the compensation value of the $d(x,t)$.

Recalling Equation (3), the position and attitude loops are coupled. Specifically, the outputs of the $x$ and $y$ are the referenced signals of $\theta$ and $\phi$. The attitude tracking errors and angular velocity tracking errors are defined as

$$z_{A,1} = A - A_r \tag{18}$$

$$z_{A,2} = \dot{A} - \alpha_2 \tag{19}$$

where $A$ is measured attitude, $A_r$ denotes the referenced position, $\alpha_2$ is the virtual control signal.

Then, the attitude controller is defined as

$$u_A \approx \dot{z}_{A,2}, \alpha_1 = -k_{A,1}z_{P,2}, \alpha_2 = -k_{A,2}z_{P,1} \tag{20}$$

where the control parameters $k_{A,1}$ and $k_{A,2}$ are positive numbers. $u_A = \begin{bmatrix} U_2 & U_3 & U_4 \end{bmatrix}^T$.

*4.2. Stability of the Quadrotor Controller*

This section takes the position control loop as an example for the stability analysis. Combining Equations (12) and (13), it is obtained that

$$\dot{z}_{P,1} = z_{P,2} - k_{p,1}z_{Q,1} - k_{p,2}\int_0^t z_{Q,1}dt \tag{21}$$

Rewriting Equation (20), one gets a state space form:

$$\dot{z}_{Q,1} = A_Q z_{Q,1} + B_Q z_{Q,2} \tag{22}$$

where $z_{Q,1} = \left[ \int_0^t z_{Q,1}dt, z_{Q,1} \right]^T$, $A_Q = \begin{bmatrix} 0 & 1 \\ -k_{p,2} & -k_{p,1} \end{bmatrix}$, and $B_Q = \begin{bmatrix} 0 & 1 \end{bmatrix}^T$.

As stated earlier in Reference [44,45], the input-to-state criterion is necessary and sufficient for stability analysis.

**Remark 3.** *For $^\exists \lambda_0 > 0$, $^\exists \rho_0 > 0$, if the virtual control signal $\alpha_1$ is incorporated into Equation (12) and the boundary of $z_{P,2}$ is uniformly defined, one gets*

$$|z_{Q,1}(t)| \leq \lambda_0 e^{-\rho_0 t}|z_{Q,1}(0)| + \frac{\lambda_0}{\rho_0}\left[ \sup_{0 \leq \tau \leq t} |z_{P,2}(\tau)| \right] \tag{23}$$

Substituting the defined control law (17) into Equation (15) yields

$$\dot{z}_{P,2} = -k_{p,3}\dot{z}_{Q,2} + d(x,t) - \hat{d}(x,t) \tag{24}$$

The solution of Equation (24) is calculated as

$$z_{P,2} = -k_{p,3}e^{-k_{p,3}t} + e^{-k_{p,3}t}\int_0^{T_s} \left( d(x,t) - (\hat{d}(x,t))e^{-k_{p,3}\tau}d\tau \right) \tag{25}$$

In practice, the disturbances $d(x,t)$ are bounded. Hence, $d(x,t) - (\hat{d}(x,t))$ are bounded. Therefore, one obtains

$$|z_{P,2}(t)| \leq \beta_1 |z_{P,2}(0)|e^{-\beta_0 t} + \beta_2 \sup_{0 \leq \tau_s \leq t} |d(x,t) - (\hat{d}(x,t))| \tag{26}$$

where $\beta_0$, $\beta_1$, and $\beta_2$ are positive numbers.

According to Equations (23) and (26), the stability of the position control loop can be guaranteed. In addition, the stability analysis of the attitude loop is the same as the position loop.

*4.3. LESO Design*

For the control of our aerial manipulator in joint space, each joint can be equated to a second-order system. Taking joint 1 $q_1$ as an example, the original second-order system under the standard consideration can be described as an integral chain system, as follows:

$$\begin{cases} \ddot{q}_1 = M_1^{-1}(\tau_1 - F_1 - \tau_{d1}) \\ y_1 = q_1 \end{cases} \tag{27}$$

where $y_1$ is the output of the second-order system.

System (27) can be changed into a state space form:

$$\begin{cases} \dot{x} = Ax + B_1 u + B_2 f_1 \\ y = Cx \end{cases} \tag{28}$$

where $x = [q_1, \dot{q}_1]^T$, $u = \tau_1$, $y = y$, $f_1 = -F_1 - \tau_{d1}$. The other matrices have the form:

$$A = \begin{bmatrix} 1 & 0 \\ 0 & 1 \end{bmatrix}, B = \begin{bmatrix} 0 \\ M_1^{-1} \end{bmatrix}, E = \begin{bmatrix} 0 \\ M_1^{-1} \end{bmatrix}, C = \begin{bmatrix} 1 & 0 \end{bmatrix} \tag{29}$$

**Remark 4.** *The external disturbance $\tau_{d1}$ and the internal unmodeled characteristics $F_1$ constitute the lumped disturbances $f_1$. $f_1$ has the property of differentiable and bounded, which satisfies $\|f_1\| < \infty$, $\|\dot{f}_1\| < \infty$, $\sup_{t>0} \|f_1\| = f_{b1}$, and $\sup_{t>0} \|\dot{f}_1\| = \dot{f}_{b1}$.*

Adding an extended state to characterize the lumped disturbances, the system (28) can be modified as

$$\begin{cases} \dot{\bar{z}} = \overline{A}\bar{z} + \overline{B}u + \overline{E}\dot{f} \\ y = \overline{C}\bar{z} \end{cases} \tag{30}$$

where $\bar{z} = [q_1, \dot{q}_1, x_3]^T$, $x_3 = f$. The other matrices have the form:

$$\overline{A} = \begin{bmatrix} 0 & 1 & 0 \\ 0 & 0 & 1 \\ 0 & 0 & 0 \end{bmatrix}, \quad \overline{B} = \begin{bmatrix} 0 \\ M_1^{-1} \\ 0 \end{bmatrix}, \quad \overline{E} = \begin{bmatrix} 0 \\ 0 \\ M_1^{-1} \end{bmatrix}, \quad \overline{C} = \begin{bmatrix} 1 & 0 & 0 \end{bmatrix} \tag{31}$$

For system (31), the LESO is defined as

$$\begin{cases} \dot{\hat{\bar{z}}} = A\hat{\bar{z}} + Bu + L(y - \hat{y}) \\ \hat{y} = C\hat{\bar{z}} \end{cases} \tag{32}$$

where $\hat{\bar{z}} = [\hat{q}_1, \hat{\dot{q}}_1, \hat{x}_3]^T$, $\hat{\bar{z}}$ is the estimation of the $\bar{z}$. $L = [\beta, \beta_2, \beta_3]^T = [\xi_1 \omega_o, \xi_2 \omega_o, \xi_3 \omega_o]^T$ is observer gain, $\omega_o > 0$ is the observer bandwidth, $\hat{y}$ is the system output. $\xi_i (i = 1, 2, 3)$ is the root of the characteristic equation $\lambda(s) = s^3 + \varsigma_1 s^2 + \varsigma_2 s + \varsigma_3$, which is described as

$$\varsigma_i = \frac{3!}{i!(3-i)!}, i = 1,2,3 \tag{33}$$

The stability analysis of the proposed observer LESO can be found in our previous works [46].

*4.4. Manipulator Controller Design*

In this section, joint 1 $q_1$ will be considered as an example to display the BIFTSMC design. For joint 1, the errors of the tracking position, velocity, and acceleration are defined as

$$\begin{cases} e_1 = q_1 - q_{1d} \\ \dot{e}_1 = \dot{q}_1 - \dot{q}_{1d} \\ \ddot{e}_1 = \ddot{q}_1 - \ddot{q}_{1d} \end{cases} \tag{34}$$

where $q_{1d}$, $\dot{q}_{1d}$, and $\ddot{q}_{1d}$ are the referenced joint position, velocity, and acceleration. $\dot{q}_1$ and $\ddot{q}_1$ are the velocity and acceleration of joint 1.

According to references [47], an IFTSM surface is defined as

$$s_1 = \int \left( \dot{e}_1 + \frac{2\alpha_1}{1 + e^{-\beta_1 |(e_1| - \phi)}} e_1 + \frac{2\alpha_2}{1 + e^{\beta_2(|e_1| - \phi)}} |e_1|^K \operatorname{sgn}(e_1) \right) \tag{35}$$

where $\alpha_1$, $\alpha_2$, $\beta_1$, and $\beta_2$ are the positive numbers. $0 < K < 1$, and $\phi = \left( \frac{\alpha_2}{\alpha_1} \right)^{1/(1+\kappa)}$.

Taking the first-order derivative and second-order derivative of the sliding mode function (35) yields

$$\dot{s}_1 = \dot{e}_1 + \frac{2\alpha_1}{1 + e^{-\beta_1(|e_1| - \phi)}} x_{ei} + \frac{2\alpha_2}{1 + e^{\beta_2(|e_1| - \phi)}} |e_1|^K \operatorname{sgn}(e_1) = 0 \tag{36}$$

$$\begin{aligned} \ddot{s}_1 = \ddot{e}_1 + &\frac{2\alpha_1}{1 + e^{-\beta_1(|e_1| - \phi)}} \dot{e}_1 + \frac{2\alpha_1\beta_1\dot{e}_1 \operatorname{sgn}(e_1) e^{-\beta_1(|e_1| - \phi)}}{\left(1 + e^{-\beta_1(|e_1| - \phi)}\right)^2} e_1 + \frac{2\alpha_2 K}{1 + e^{\beta_2(e_1| - \phi)}} |e_1|^{K-1} \dot{e}_1 \\ &- \frac{2\alpha_2\beta_2 e_1 e^{\beta_2(|e_1| - \phi)}}{\left(1 + e^{\beta_2(e_1| - \phi)}\right)^2} |e_1|^K \end{aligned} \tag{37}$$

Then, the Equation (37) can be simplified as

$$\ddot{s}_1 = \ddot{e}_1 + Z_1 \tag{38}$$

$$\begin{aligned} Z_1 = &\frac{2\alpha_1}{1 + e^{-\beta_1(|\varrho| - \phi)} \dot{e}_1} + \frac{2\alpha_1\beta_1\dot{e}_1 \operatorname{sgn}(e_1) e^{-\beta_1(|\beta_1| - \phi)}}{\left(1 + e^{-\beta_1(|\rho_1| - \phi)}\right)^2} e_1 + \frac{2\alpha_2 K}{1 + e^{\beta_2(|\rho_1| - \phi)}} |e_1|^{K-1} \dot{e}_1 \\ &- \frac{2\alpha_2\beta_2 e_1 e^{\beta_2(|\rho_1| - \phi)}}{\left(1 + e^{\beta_2(|\beta_1| - \phi)}\right)^2} |e_1|^K \end{aligned} \tag{39}$$

Combining Equations (5), (36), and (37), one obtains the third-order state space model as

$$\begin{cases} \dot{s}_1 = s_2 \\ \dot{s}_2 = s_3 \\ \dot{s}_3 = \frac{d}{dt}\left( M_1^{-1}(\tau_1 - H_1) - \ddot{q}_{1d} + Z_1 \right) \end{cases} \tag{40}$$

A backstepping design approach is suggested to attain the effective control torque for the dynamic system as specified in Equation (22). To achieve this, a change of coordinate is initiated as follows:

$$\begin{cases} Y_1 = s_1 \\ Y_2 = s_2 - \delta_1 \\ Y_3 = s_3 - \delta_2 \end{cases} \tag{41}$$

Furthermore, the demonstration of global asymptotic stability for the formulated control strategy will be carried out through a sequential process consisting of three steps.

Step 1: a Lyapunov function is chosen as

$$V_1 = 0.5 Y_1^T Y_1 \tag{42}$$

where $Y_1$ is the intermediate variable.

Taking the derivative of $Y_1$ with respect to time gives

$$\dot{Y}_1 = \dot{s}_1 = s_2 = Y_2 + \delta_1 \tag{43}$$

where $\delta_1$ is the virtual control input. $Y_2$ is the intermediate variable.

Taking the derivative of Equation (41), one gets

$$\dot{V}_1 = Y_1^T \dot{Y}_1 = Y_1^T (Y_2 + \delta_1) \tag{44}$$

The virtual control $\delta_1$ is chosen in a suitable manner to ensure that the first-order system can be stabilized according to the following equation:

$$\delta_1 = -\xi_1 Y_1 \tag{45}$$

where $\xi_1$ is a positive number, which is the control parameter.

Substituting Equation (29) into Equation (28), one gets

$$\dot{V}_1 = Y_1 Y_2 - \xi_1 Y_{1i}^2 \tag{46}$$

From Equation (45), it can be observed that if $Y_2 = 0$, then $\dot{V}_1 = -\xi_1 Y_{1i}^2 \leq 0$. Therefore, $Y_1$ will be asymptotically stable.

Step 2: considering another Lyapunov function as

$$V_2 = V_1 + 0.5 Y_2^T Y_2 \tag{47}$$

Derivating for $Y_2$ in Equation (41) yields

$$\dot{Y}_2 = \dot{s}_2 - \dot{\delta}_1 = Y_3 + \delta_2 - \dot{\delta}_1 == Y_3 + \delta_2 + \xi_1 s_2 \tag{48}$$

Derivating for $V_2$ in Equation (47) and combining Equation (48) yields

$$\dot{V}_2 = \dot{V}_1 + Y_2^T \dot{Y}_2 = -\xi_1 Y_1^2 + Y_1^T Y_2 + Y_2^T (Y_3 + \delta_2 + \xi_1 s_2) \tag{49}$$

Then, the virtual control law is selected to eliminate the intermediate variable:

$$\delta_2 = -Y_1 - \varepsilon_2 Y_2 - \varepsilon_1 S_2 \tag{50}$$

Substituting Equation (49) into Equation (50) yields

$$\begin{aligned} \dot{V}_2 &= -\xi_1 Y_1^2 + Y_1^T Y_2 + Y_2^T (Y_3 - Y_1 - \xi_2 Y_2 - \xi_1 s_2 + \xi_1 s_2) \\ &= -\xi_1 Y_1^2 - \xi_2 Y_2^2 + Y_2^T Y_3 \end{aligned} \tag{51}$$

When $Y_3 = 0$, then $\dot{V}_2 = -\xi_1 Y_1^2 - \xi_2 Y_2^2 + Y_2^T \leq 0$. In this case, the states $Y_1$ and $Y_2$ will be asymptotically stable.

Step 3: Consider another Lyapunov function as follows:

$$V_3 = V_2 + 0.5 Y_3^T Y_3 \tag{52}$$

Derivating for $V_3$ in Equation (52) yields

$$\dot{V}_3 = \dot{V}_2 + Y_3^T \dot{Y}_3 \tag{53}$$

Substituting Equations (40), (41), and (51) into Equation (53), one obtains

$$
\begin{aligned}
\dot{V}_3 = & -\xi_1 Y_1^2 - \xi_2 Y_2^2 + Y_2^T Y_3 \\
& + Y_3^T \left( \frac{d}{dt} \left( M_1^{-1}(\tau_1 - H_1) - \ddot{q}_{1d} + Z_1 \right) - \dot{\delta}_2 \right)
\end{aligned}
\tag{54}
$$

Hence, the BIFTSMC can be defined as

$$u = u_{eq} + u_{sw} \tag{55}$$

$$u_{eq} = M_1(\ddot{q}_{1d} - Z_1 + \delta_2) + \hat{x}_3 - M_1 \int (\varepsilon_3 Y_3 + Y_2) \tag{56}$$

$$\dot{u}_{sw} = \left( \Xi + \dot{f}_{b1} \right) \text{sign}(Y_3) \tag{57}$$

where $u_{eq}$ is the equivalent control law, $u_{sw}$ is the reaching control law, $\Xi > 0$ is a small constant.

Substituting the defined controller into Equation (37), one gets

$$
\begin{aligned}
\dot{V}_3 = & -\xi_1 Y_1^2 - \xi_2 Y_2^2 - \xi_3 Y_3^2 + Y_3^T \frac{d}{dt}(\hat{x}_3 - u_{sw}) \\
= & -\xi_1 Y_1^2 - \xi_2 Y_2^2 - \xi_3 Y_3^2 - \left( \Xi + \dot{f}_{b1} \right) Y_3^T \text{sign}(Y_3) + \text{sign}(Y_3) \dot{\hat{x}}_3 \\
\leq & -\xi_1 Y_1^2 - \xi_2 Y_2^2 - \xi_3 Y_3^2 - \left( \Xi + \dot{f}_{b1} \right) \left| Y_3^T \right| + \dot{f}_{b1} \left| Y_3^T \right| \\
\leq & -\xi_1 Y_1^2 - \xi_2 Y_2^2 - \xi_3 Y_3^2 - \Xi \left| Y_3^T \right| \leq 0
\end{aligned}
\tag{58}
$$

From Equation (58), the $Y_1$, $Y_2$, and $Y_3$ will reach a state of zero convergence. The controller design of the joint 2 is the same as that of joint 1.

## 5. Simulation and Results

As shown in Figure 5, the cable-driven aerial manipulator designed using Solidworks software was imported into Matlab/Simscape to verify the performance of the designed controller. In addition, the physical parameters of the designed cable-driven aerial manipulator are shown in Table 1. Next, this section will verify the performance of the designed controller through three simulation cases. It should be noted that in the aerial manipulator system, BC-DOB is used to control the quadrotor, and BIFTSMC-LESO is used to design the robotic arm. In cases 1 and 2, the quadrotor is controlled in hover mode. In case 3, the quadrotor needs to carry the robotic manipulator to complete a specific trajectory tracking control.

**Table 1.** Physical parameters of the designed cable-driven aerial manipulator.

| Parameter | Value | Explanation | Parameter | Value | Explanation |
|---|---|---|---|---|---|
| $m$ | 3 kg | Mass of the quadrotor | $m_1$ | 0.06 kg | Mass of the link 1 |
| $m_2$ | 0.225 kg | Mass of link 2 | g | 9.8 m/s$^2$ | Gravitational acceleration |
| $l_1$ | 0.05 m | Length of link 1 | $l_2$ | 0.15 m | Length of link 2 |
| $J_{xx}$ | 0.287 kg·m$^2$ | Rotational inertia of the quadrotor around the *x*-axis | $J_{yy}$ | 0.314 kg·m$^2$ | Rotational inertia of the quadrotor around the *y*-axis |
| $J_{zz}$ | 0.1477 kg·m$^2$ | Rotational inertia of the quadrotor around the *z*-axis | $I_{m1}$ | 0.102 kg·m$^2$ | Inertia of motor 1 |
| $I_{m2}$ | 0.811 kg·m$^2$ | Inertia of motor 2 | $D_{m1}$ | 0.001 kg·m$^2$ | Damp of motor 1 |
| $D_{m2}$ | 0.001 kg·m$^2$ | Damp of motor 2 | $L$ | 0.630 m | Distance between the rotation axes and center of quadrotor |
| $k_t$ | $1.1719 \times 10^{-5}$ | Thrust coefficient | $k_m$ | $0.198 \times 10^{-5}$ | Torque coefficient |

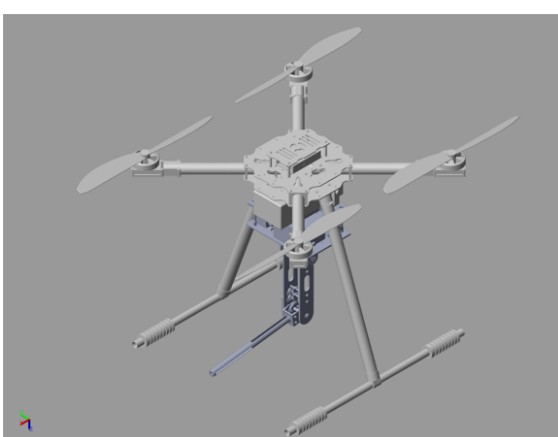

**Figure 5.** Aerial manipulator in Matlab/simscape.

*5.1. Case 1*

The purpose of this case is to tune the parameters of the designed controller. The referenced joint positions are set as $q_{1r} = 60°$ and $q_{2r} = -60°$. The joint angular velocity and angular acceleration of the manipulator are set to 0, and the simulation time lasting 10 s with 50 Hz sampling time. Meanwhile, two other controllers are considered, including the SMC-ESO (sliding mode based on extended state observer) in the reference [46], and the LADRC (linear active disturbance rejection controller) proposed in the reference [48]. These controllers are introduced as comparisons to investigate the performance of the proposed controller, and are referred to as controller 1 and controller 2, respectively. In addition, Gaussian noise signals with mean 0 and variance 0.01 are added to the manipulator dynamics model and measurement terminal, respectively. The parameters of all controllers are tuned by the improved salp swarm algorithm (ISSA). More details can be found in our previous work [49]. The optimized control parameters of the three controllers by ISSA are listed in Tables 2–4.

**Table 2.** The control parameters of the proposed controller tuned by ISSA.

| Parameter | Value | Parameter | Value |
|---|---|---|---|
| $\omega_{o1}$ | 401 | $\omega_{o2}$ | 380 |
| $\alpha_{11}$ | 2 | $\alpha_{21}$ | 1 |
| $\alpha_{12}$ | 3 | $\alpha_{22}$ | 2 |
| $\beta_{11}$ | 1 | $\beta_{12}$ | 1 |
| $\beta_{12}$ | 1 | $\beta_{22}$ | 1 |
| $k_{11}$ | 0.5 | $k_{21}$ | 0.5 |

**Table 3.** The control parameters of SMC-ESO tuned by ISSA.

| Parameter | Value | Parameter | Value |
| --- | --- | --- | --- |
| $\beta_{11}$ | 120 | $\beta_{21}$ | 96 |
| $\beta_{12}$ | 303 | $\beta_{22}$ | 316 |
| $\beta_{31}$ | 800 | $\beta_{32}$ | 780 |
| $\alpha_{11}$ | 0.5 | $\alpha_{21}$ | 0.5 |
| $\alpha_{12}$ | 0.5 | $\beta_{22}$ | 0.5 |
| $\delta_1$ | 0.02 | $\delta_2$ | 0.02 |
| $k_1$ | 87 | $k_2$ | 79 |
| $c_1$ | 1.5 | $c_2$ | 1.5 |

**Table 4.** The control parameters of LADRC tuned by ISSA.

| Parameter | Value | Parameter | Value |
| --- | --- | --- | --- |
| $\omega_{o1}$ | 500 | $\omega_{o2}$ | 500 |
| $\omega_{c1}$ | 39 | $\omega_{c2}$ | 45 |

The simulation results are presented in Figures 6–10. From Figures 6 and 7, it can be observed that all three controllers are able to track the reference trajectory well. Specifically, all three controllers exhibit good transient performance (such as short rise time, almost no overshoot, and reduced settling time) and steady-state performance (such as high tracking accuracy and strong disturbance rejection capability). As can be seen from the local zoomed-in graph, the proposed controller has better control performance compared to the other two controllers. For the response of joint 2, although this controller has a slightly larger overshoot, it is still within the acceptable range ($\leq 5\%$).

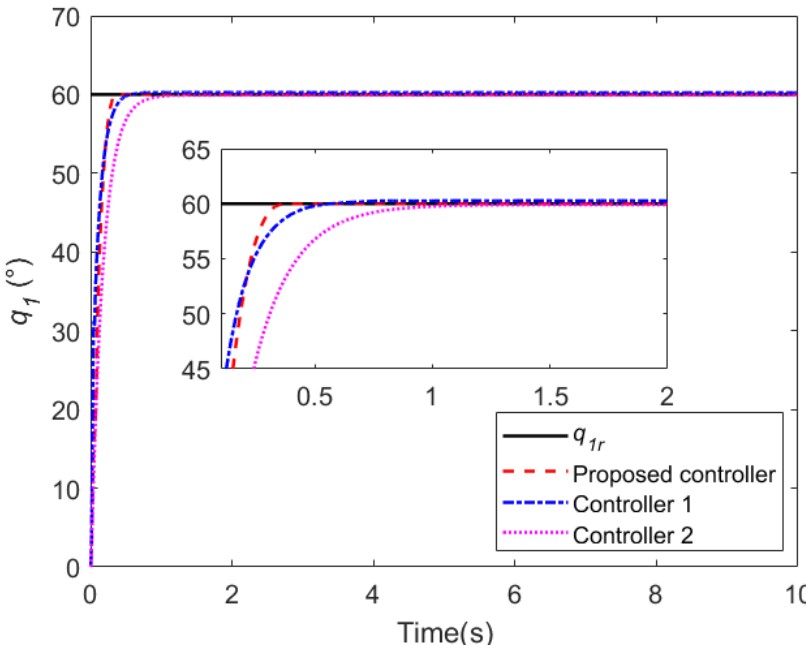

**Figure 6.** Response of joint 1 in case 1.

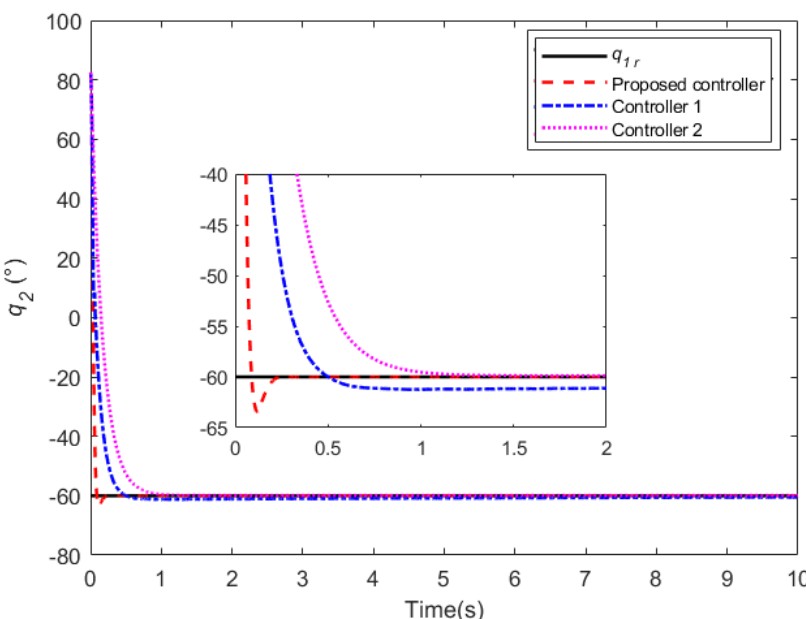

**Figure 7.** Response of joint 2 in case 1.

The trajectory tracking errors of joint 1 and joint 2 are given in Figures 8 and 9. In terms of time scale, the time to steady state of the proposed controller is 33.3% and 55.6% faster than that of controller 1 and controller 2, respectively. In terms of convergence speed, the convergence speed of the proposed controller is significantly higher than the other two controllers. The role of case 1 is to offer a set of applicable parameters for other simulation cases.

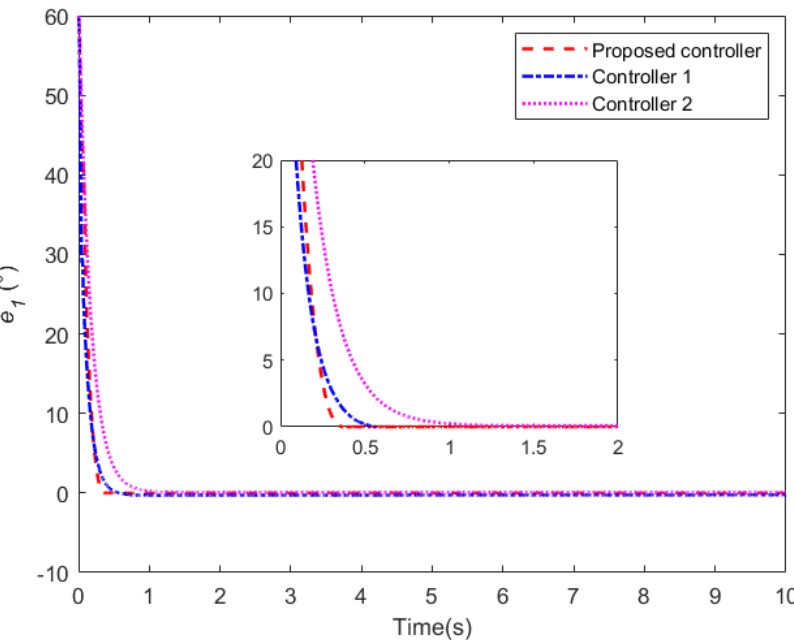

**Figure 8.** Tracking error of joint 1 in case 1.

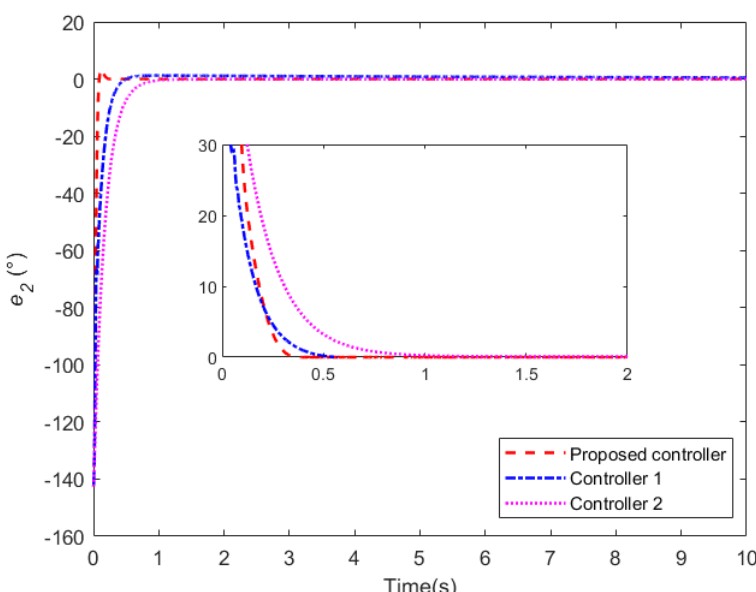

**Figure 9.** Tracking error of joint 2 in case 1.

The estimation results of the two state observers for the lumped disturbance are given in Figure 10. It can be seen from the figure that both ESO and LESO can effectively estimate the disturbances, and the estimation accuracy of LESO is weaker than that of ESO, but the estimation time is less than that of LESO. This may be related to the structure of the two observers.

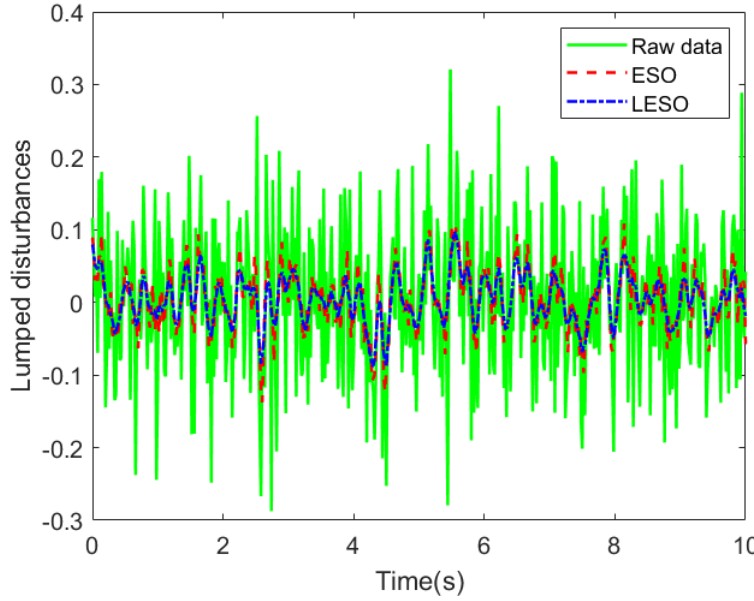

**Figure 10.** Comparison of ESO and LESO in case 1.

Furthermore, a comparison of the anti-interference ability of the three controllers under different disturbances has been analyzed. The robustness value is used to express the anti-interference ability to disturbances, which is calculated as

$$
\text{Robustness} = \frac{\sum_{i=1}^{N}\left(f_i - \bar{f}\right)\left(\hat{f}_i - \overline{\hat{f}}\right)}{\sqrt{\sum_{i=1}^{N}\left(f_i - \bar{f}\right)^2 \sum_{i=1}^{N}\left(\hat{f}_i - \overline{\hat{f}}\right)^2}} \times 100\%
\tag{59}
$$

where $\bar{f}$ denotes the mean value of the raw disturbances $f_i$, $\hat{f}$ denotes the mean value of the estimated disturbances $\hat{f}_i$. The larger the robustness value, the more robust the controller is to disturbances.

In the simulation, the robustness of the three controllers is tested by varying the variance of the Gaussian noise. As can be seen from Figure 11, the robustness of the controller in this paper is higher than the other two controllers when the disturbance variance is in the range of 0.001 to 1. When the disturbance variance is higher than 1, the robustness of this controller decreases faster than the other two controllers. Further, when the disturbance variance is less than 1, LESO has better disturbance compensation ability than ESO, which indicates that the proposed controller is more suitable for processing disturbances with smaller amplitude.

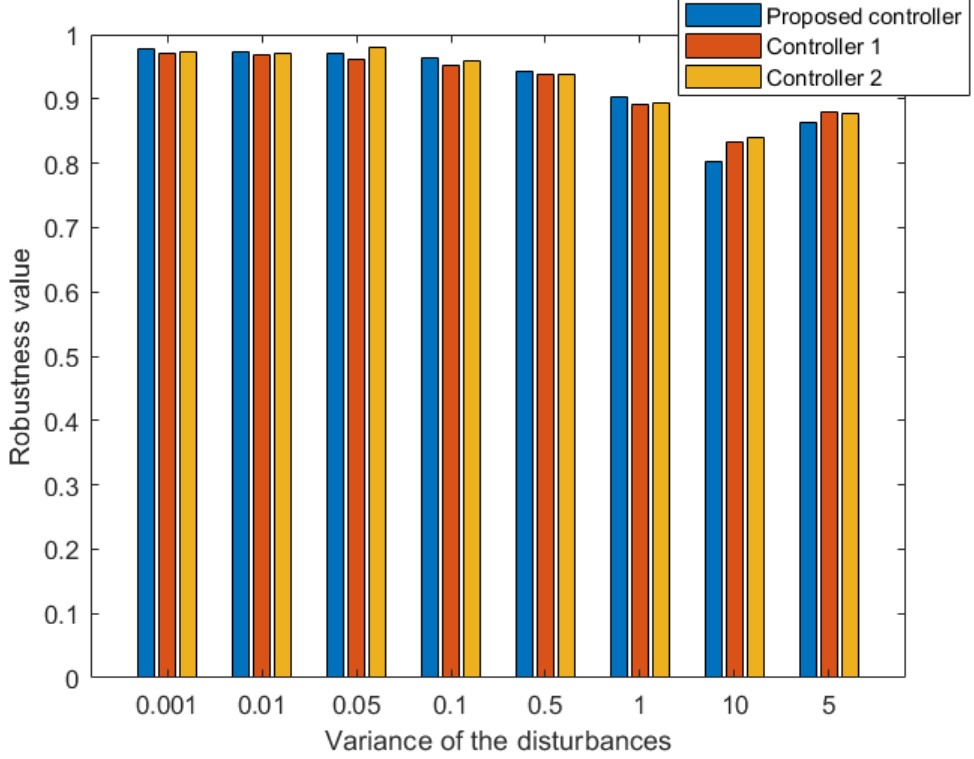

**Figure 11.** Robustness of the three controllers in case 1.

### 5.2. Case 2

The purpose of this case is to test the disturbance rejection performance of the proposed controller. The referenced trajectories of joint 1 and joint 2 are defined as $q_{1r} = 55 \sin(0.4\pi t)$ and $q_{2r} = 36 \sin(0.4\pi t)$. The joint angular velocity and angular acceleration of the manipulator are set to 0, and the simulation time is 10 s. In addition, Gaussian noise signals with mean 0 and variance 0.01 are added to the manipulator dynamics model and measurement terminal, respectively. The LADRC and SMC-ESO are also considered as comparisons.

All three controllers can guarantee strong control performance under high nonlinearities and time-varying disturbances, as illustrated in Figures 12 and 13, which proves the effectiveness of LESO or ESO. The proposed controller performs the best control performance among all three controllers, as illustrated in Figures 14 and 15, and has the least tracking errors. Furthermore, the two indicators, named max mean error (MME) and root mean square error(RMSE), are introduced to evaluate the tracking errors of the three controllers. As shown in Figures 16 and 17, the MME and RSME of the proposed controller are the smallest. Taking joint 1 as an example, the MME of the proposed controller is 0.3828, which is 98.23% and 86.29% of the other controllers, respectively. The RSME of the proposed controller is 0.6612, which is 87.68% and 89.7% of the other controllers, respectively. All these results show that the

controller developed in this paper has good perturbation rejection capability, and can observe nonlinear and time-varying perturbations quickly and accurately. Furthermore, the proposed controller and controller 1 in this paper have a smaller tracking error compared to controller 2. This is owing to the fact that LESO can estimate the disturbances faster and better due to its simpler structure than ESO.

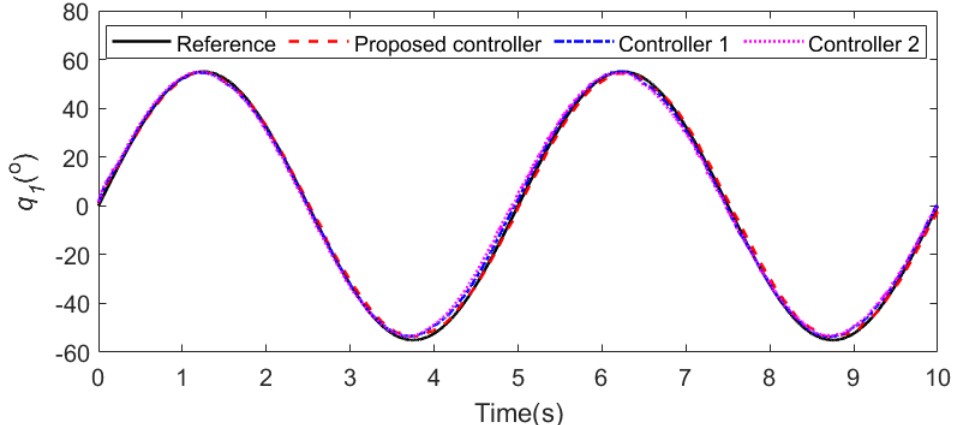

**Figure 12.** Response of joint 1 in case 2.

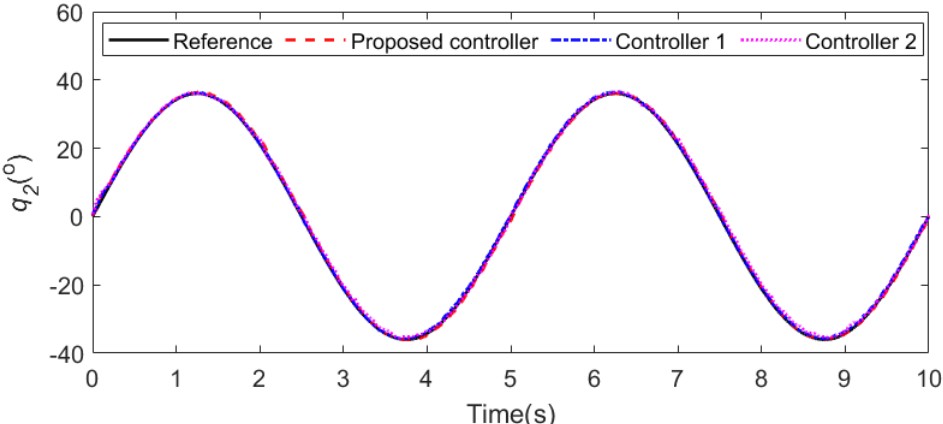

**Figure 13.** Response of joint 2 in case 2.

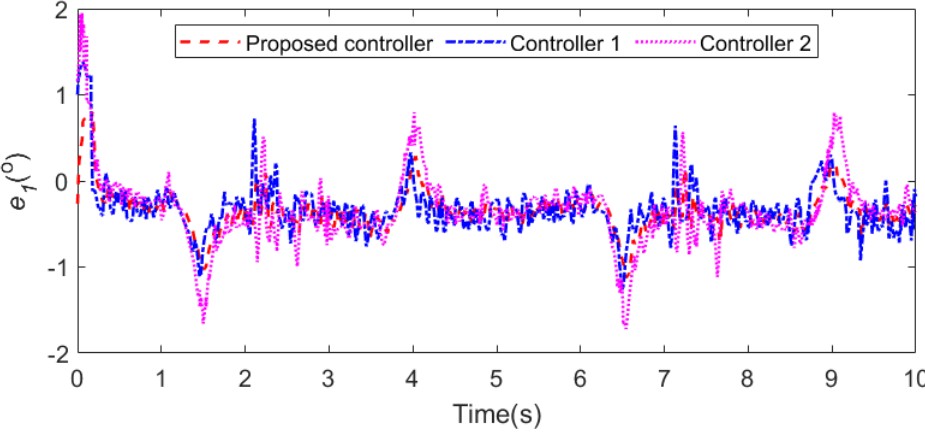

**Figure 14.** Tracking error of joint 1 in case 2.

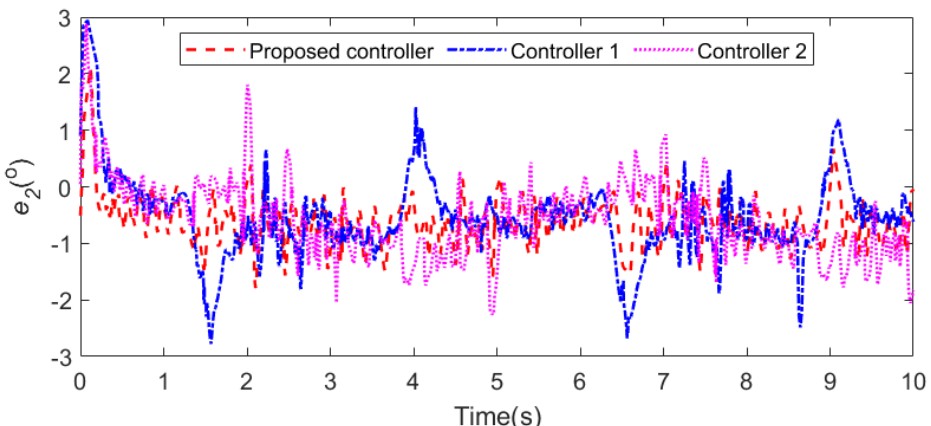

**Figure 15.** Tracking error of joint 2 in case 2.

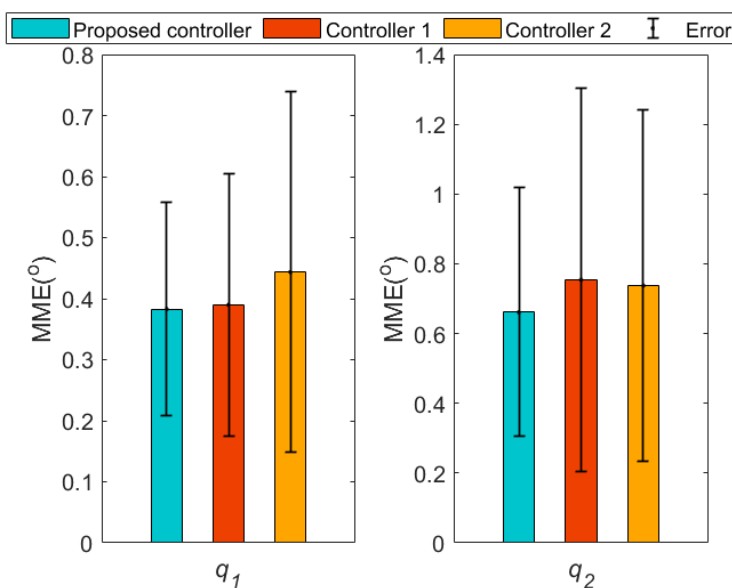

**Figure 16.** Max mean error of the three controllers.

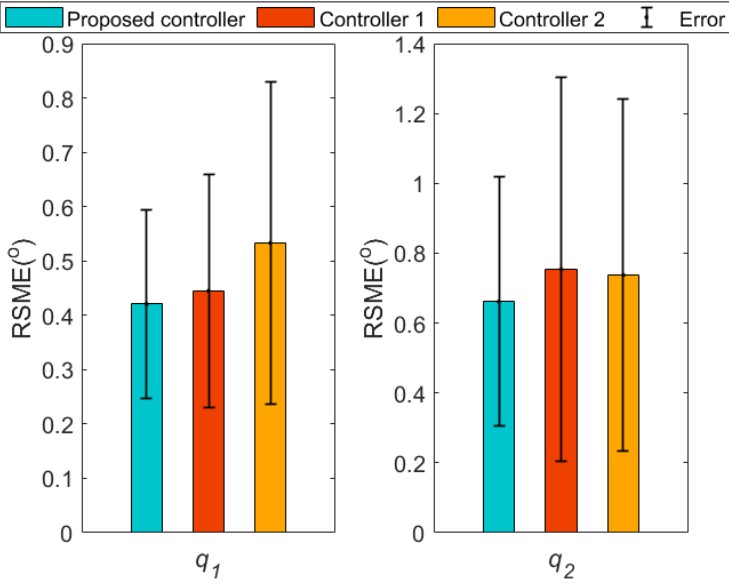

**Figure 17.** Root mean square error of the three controllers.

The control torque signals of joint 1 and joint 2 under the three controllers are shown in Figures 18 and 19. As can be seen from the graphs, the proposed controller is relatively smooth, especially during joint commutation (3~4 s, 6~7 s, and 9~10 s). Taking joint 1 as an example, when the tracking error is about 1° (2~2.5 s), the boundary layer thickness of the proposed controller increases, which enables the state to converge to the sliding mode surface with a faster convergence rate. At the same time, the chattering phenomenon is effectively weakened, while the output torques of the other two controllers at this stage exhibit stronger chattering. The control torque of controller 2 also undergoes excitation, which can only pull the system state volume back to the target value by the amount of excitation torque. Therefore, the performance of the proposed controller is better than that of the other two controllers.

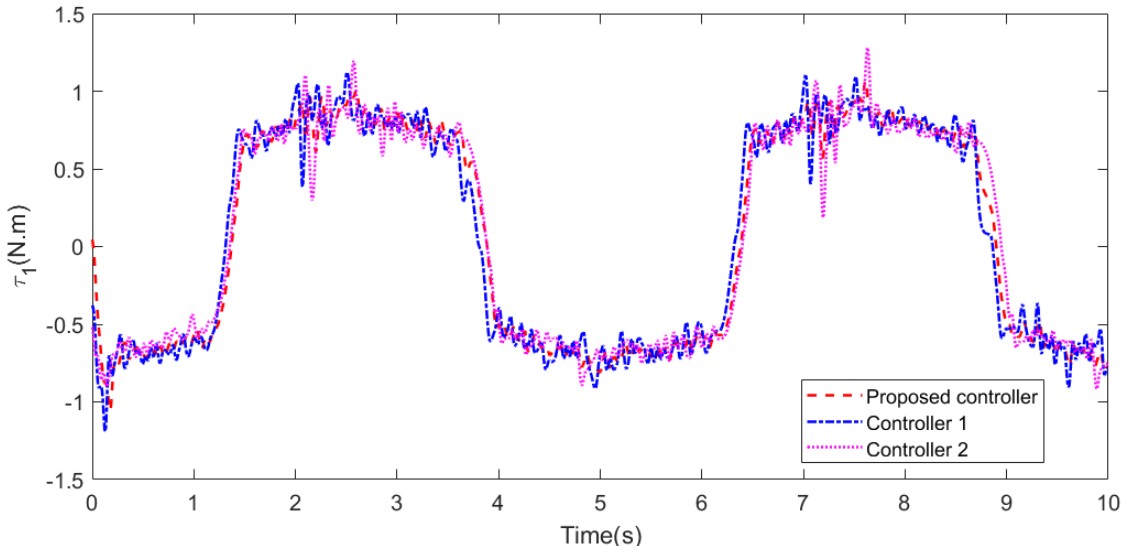

**Figure 18.** Torque response of joint 1 in case 2.

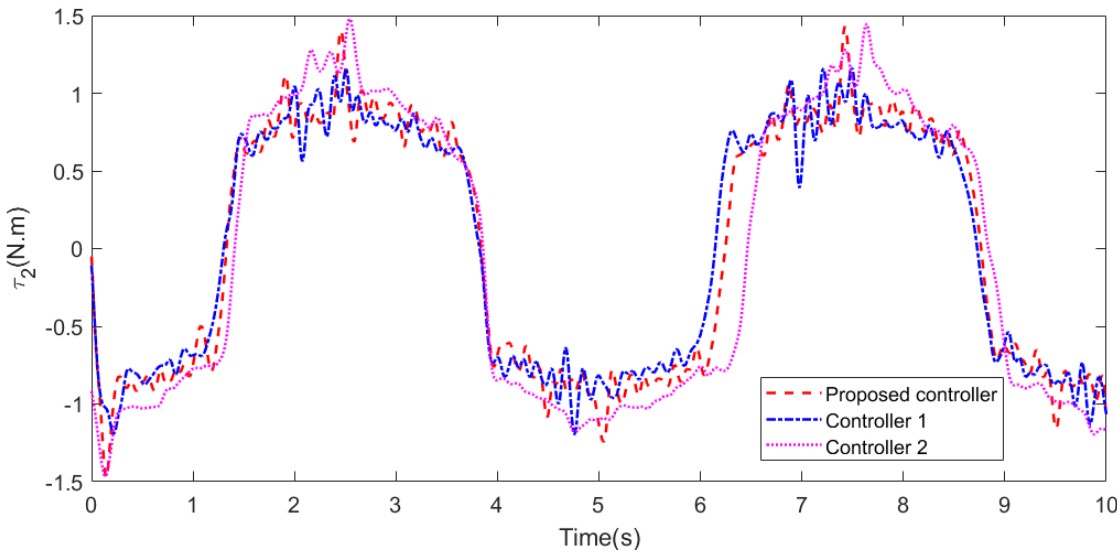

**Figure 19.** Torque response of joint 2 in case 2.

*5.3. Case 3*

The process of water sampling by the UAM is simulated in this case. In the simulation, a particular flight trajectory is designed. Firstly, the UAM from the origin $(0,0,0)m$ of takeoff, after 4 s, reaches the height peak $(4,5,3)m$. Then, the UAM arrives at the water quality sampling point $(6,8,1.5)m$ and hovers after 2 m, and the sampling time is 10 m. Furthermore,

the UAM flies to the other height peak $(8,10,1.5)m$ after 2 s. Lastly, the UAM lands at the endpoint $(14,20,0)m$ after 4 s. During 6∼16 s, the cable-driven manipulator is driven to draw water. Due to the slow time-varying characteristics of wind gusts, Gaussian noise signals with mean 0 and variance 0.1 are added to simulate them. Meanwhile, random noise signals with mean 0 and variance 0.001 are added to the manipulator dynamics model and measurement terminal, respectively. The joint angular velocity and angular acceleration of the manipulator are set to 0, and the linear velocity, linear acceleration, angular velocity, and angular acceleration of the quadrotor are set to 0. In addition, the control parameters of the quadrotor are listed in Table 5. These parameters are also tuned by ISSA.

**Table 5.** The parameters of the quadrotor controller.

| Parameter | Value |
| :---: | :---: |
| $k_{p,1}$ | $[5,5,2]^T$ |
| $k_{p,2}$ | $[0.02,0.02,0.01]^T$ |
| $k_{p,3}$ | $[10,10,8]^T$ |
| $k_{A,1}$ | $[6,7,1]^T$ |
| $k_{A,2}$ | $[1,1,0.5]^T$ |

Figure 20 shows the 3D motion trajectories of the UAM in the inertial coordinate system, and it can be observed that the UAM accomplishes the mission better and achieves the whole process from takeoff, water sampling, and landing. The response curves of the position and attitude of the quadrotor are given in Figures 21 and 22 . It can be seen that the quadrotor can track the referenced trajectories better under the proposed controller in this paper. Further, Figure 23 presents the three-axis position tracking error of the quadrotor, and it can be seen that the position error of the $x$-axis is controlled between $-0.472∼0.469$ m, the error of the $y$-axis is controlled between $-0.462∼0.481$ m, and the position tracking error of the $z$-axis is controlled between $-0.007∼0.008$ m.

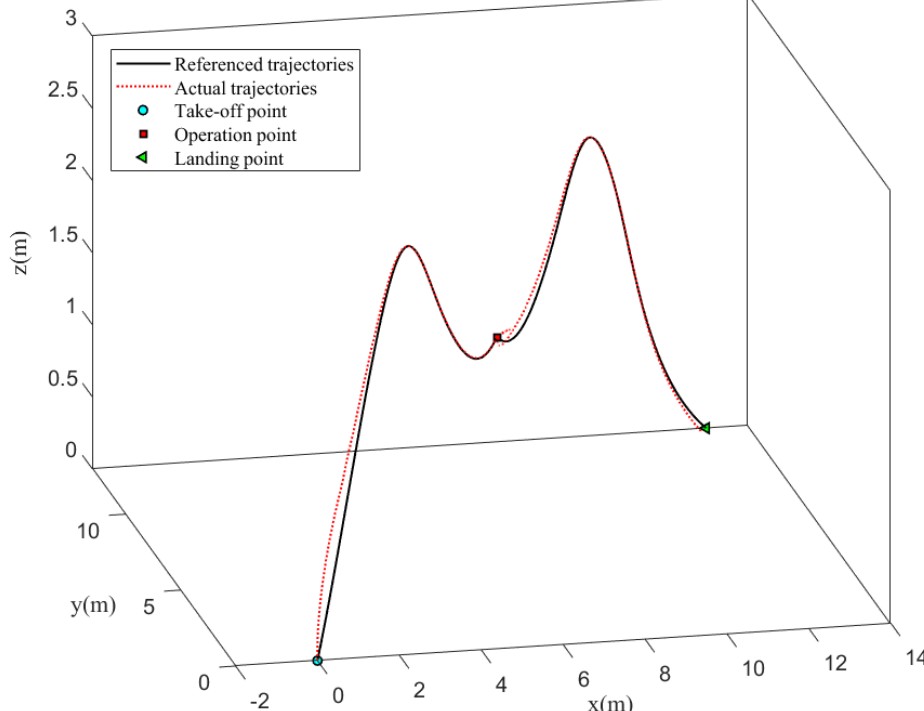

**Figure 20.** The 3D trajectories of the UAM.

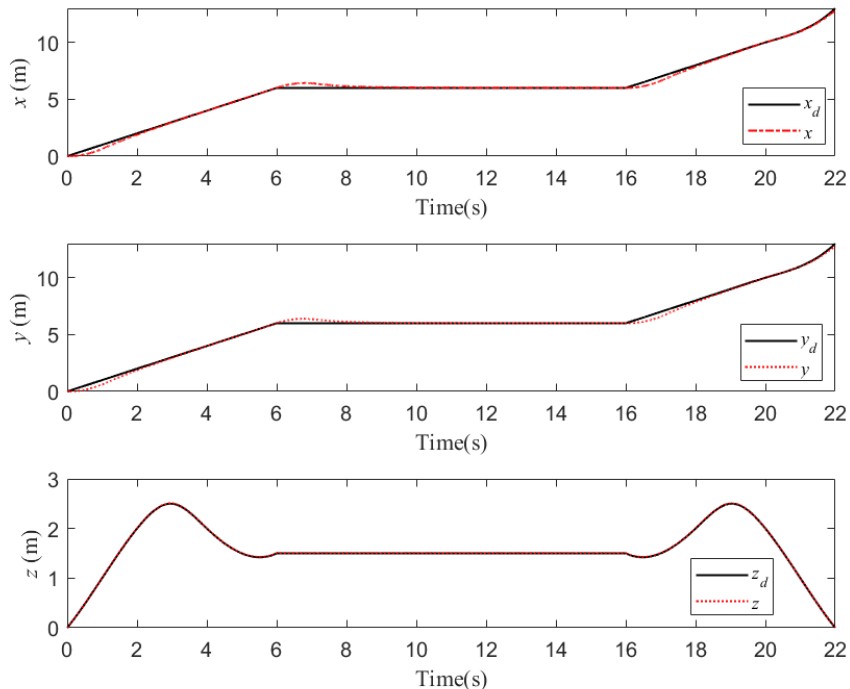

**Figure 21.** The UAM trajectories of three axes.

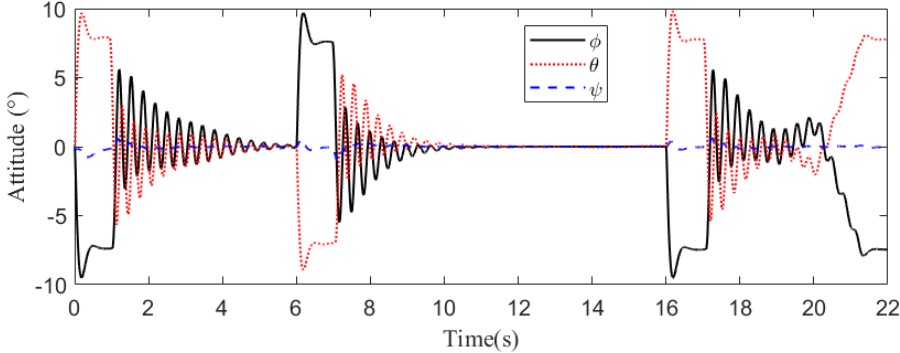

**Figure 22.** Attitude response of the UAM in case 3.

During water sampling, the initial joint angles of the cable-driven manipulator are 0° and 82.6°, respectively, and other initial conditions are 0. The cycloidal curves [50] are introduced to determine the referenced trajectories of the two joint angles, with joint 1 changing from 0° to 60° in 6∼9 s and lasting for 6 s, and then returning from 60° to 0° in 14∼16 s. Joint 2 changes from 82.6° to 0° in 6∼9 s and lasts for 6 s, and then returns from 0° to 82.6° in 14∼16 s. In addition, the same disturbances as those in case 2 are introduced into the manipulator system. The simulation results are shown in Figures 24 and 25. From Figure 24, it can be seen that the two joints can track the referenced trajectories better under the proposed controller. The disturbances are well suppressed owing to the LESO. Meanwhile, the control torques generated by the proposed controller are flatter and cause less damage to the actuators observed in Figure 25. In summary, the controller designed in this paper can better help the UAM to complete the task of water quality sampling.

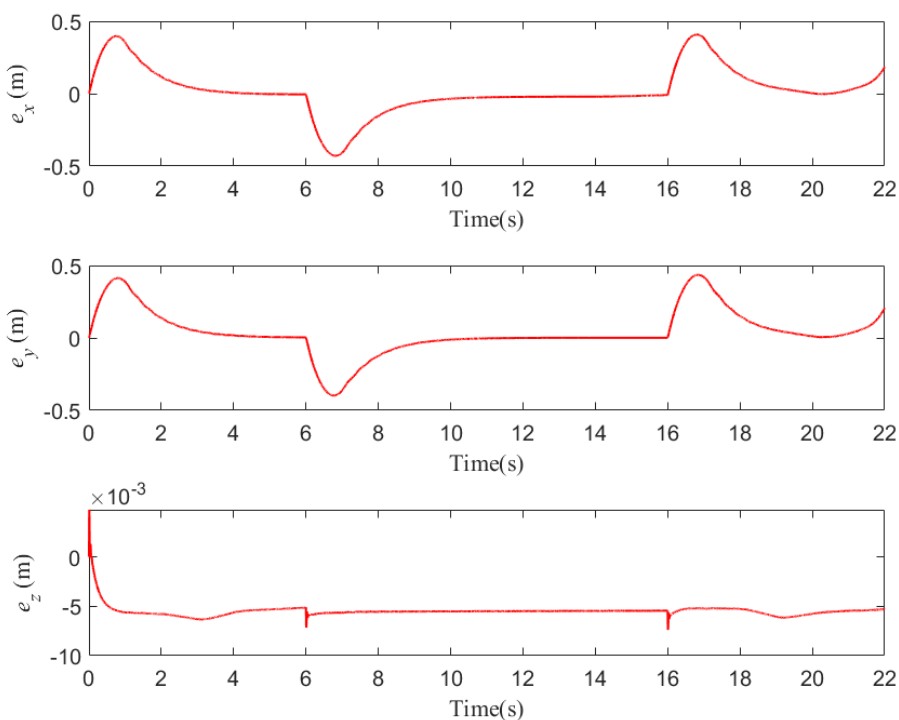

**Figure 23.** The UAM trajectory tracking errors of three axes.

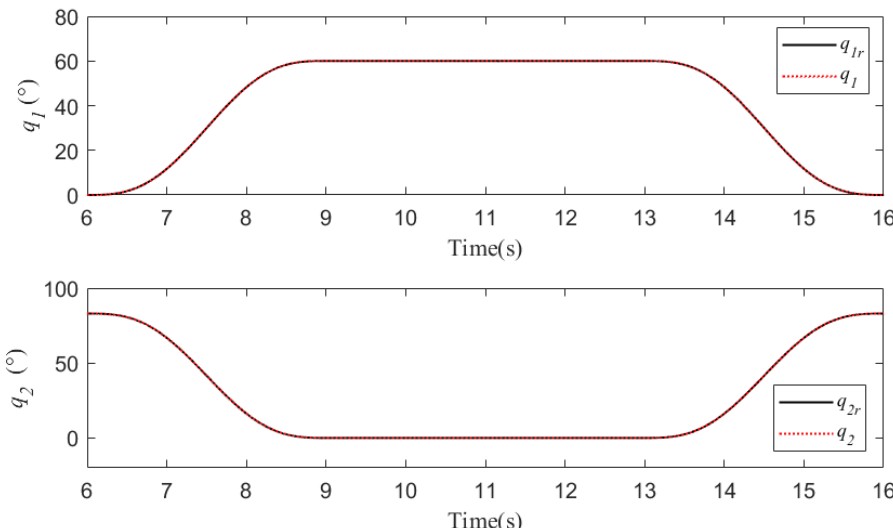

**Figure 24.** Joint response in case 3.

It should be noted that this paper does not yet have the capability to write the designed controller BIFTSMC-LESO into the hardware. Second, the actual water quality sampling experiments also need to add the vision module to obtain the position information between the aerial manipulator and the water sample. The visual positioning algorithm is also the next factor to be studied. Finally, the completion of water quality sampling experiments also requires one to obtain the authorization of the local environmental protection department. In future research, we will try to verify the effectiveness of the control algorithm proposed in this paper in practice.

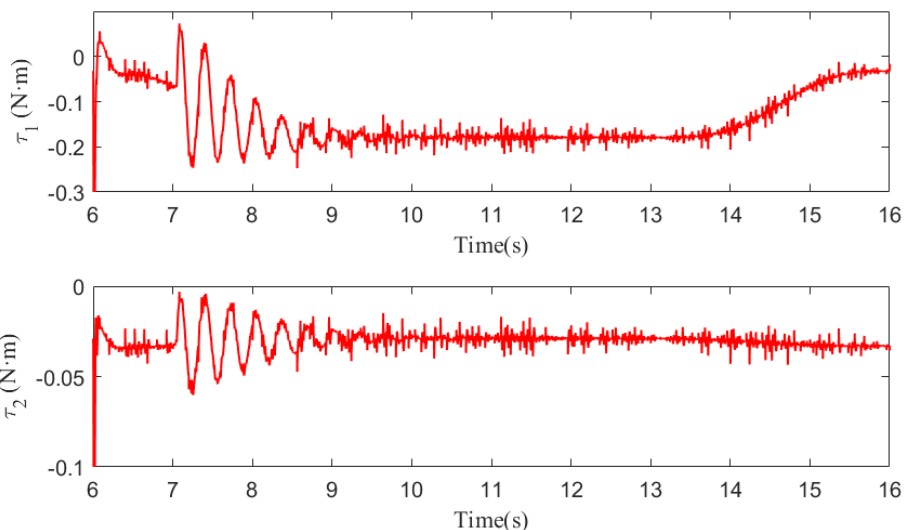

**Figure 25.** Torque response in case 3.

## 6. Conclusions

In this paper, we developed a light cable-driven aerial manipulator for water sampling. Firstly, the proposed robot system was described and designed, including the quadrotor, manipulator, cable-driven mechanism, and other lightweight mechanical constructions. Then, the system model containing kinematics and dynamics of the UAM were established and analyzed, where the Newton–Euler method was adopted to model the position dynamics and attitude dynamics of the quadrotor, and the Lagrangian method was used to deduce the manipulator dynamics with flexible joints. Especially, the external disturbances and model uncertainty are considered in the system model. Furthermore, two controllers were developed to ensure the accurate operation for the UAM. The simulation results are summarized as follows. Firstly, the BC-DOB controller enables the quadrotor to maintain position and attitude stability, allowing it to achieve high trajectory tracking control accuracy. Secondly, the BIFTSMC-LESO controller can ensure greater overall performance than LADRC or the conventional SMC-ESO by increasing the convergence speed near the equilibrium point. Thirdly, the controller parameters can be tuned by an improved salp swarm algorithm, which ensures that the controllers have good transient performance and steady-state performance. Lastly, the proposed composite controller enables the UAM to perform the water sampling task better.

In the future, we will test the feasibility of the designed controller in a real environment. Further research will focus on other aerial tasks for the UAM, such as cooperative operation, aerial inspection, and grasping.

**Author Contributions:** Conceptualization, L.D.; methodology, L.D.; software, G.Z.; validation, Y.W.; formal analysis, L.D.; investigation, Y.L.; resources, L.D.; data curation, L.D.; writing—original draft preparation, L.D.; writing—review and editing, Y.L.; visualization, L.D.; supervision, L.D.; project administration, Y.W.; funding acquisition, G.Z. All authors have read and agreed to the published version of the manuscript.

**Funding:** This paper was supported by the National Natural Science Foundation of China (No. 52005231 and No. 52175097), Social Development Science and Technology Support Project of Changzhou (No. CE20215050), and Jiangsu University Youth and Blue Project Funding.

**Data Availability Statement:** The data of simulation can be provided if necessary.

**Acknowledgments:** Thanks to the anonymous reviewers for their helpful and insightful remarks. In addition, helpful discussions with Xiaofeng Liu from Hohai University, and his guidance concerning aircraft designation, are gratefully acknowledged.

**Conflicts of Interest:** The authors declare no conflict of interest.

**Abbreviations**

The following abbreviations are used in this manuscript:

| | |
|---|---|
| AFONTSMC-NDOB | Adaptive fractional-order nonsingular terminal sliding mode based on nonlinear disturbance observer |
| BC | Backstepping control |
| BIFTSMC-LESO | Backstepping integral fast terminal sliding mode control based on linear extended state observer |
| BC-DOB | Backstepping based on disturbance observer |
| DOF | Degree-of-freedom |
| DOB | Disturbance observer |
| GPS | Global position system |
| ITSMC | Integral terminal sliding mode control |
| ISSA | Improved salp swarm algorithm |
| LESO | Linear extended state observer |
| LARC | Linear disturbance rejection controller |
| PD | Proportion derivative |
| PID | Proportion integral derivative |
| TSMC | Terminal sliding mode control |
| SMC | Sliding mode control |
| SMC-ESO | Sliding mode control based on extended state observer |
| UAM | Unmanned aerial manipulator |

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
