# Peer review of "Cable-Driven Unmanned Aerial Manipulator Systems for Water Sampling: Design, Modeling, and Control"

_drones, doi:10.3390/drones7070450_

Round 1
Reviewer 1 Report
please consider my advices

ok
Author Response
Many thanks to the experts for their careful review of this paper. We have revised the paper according to the comments of the 5 experts, please check them.

Reviewer 2 Report
This paper develops an unmanned aerial manipulator for water sampling. The study contains structure design, system modelling, and motion control. This work has a certain novelty. However, there are several suggestions to be considered before it is published.
1. Is it appropriate to simplify the quadrotor dynamics model into the form of Equation (3)? I think this is too idealistic.
2. Since the controller designed in this paper is model-free, why is it necessary to give the physical parameters of the system in Table 1.
3. In the simulation, the optimized control parameters of the other controllers should be listed.
4. In case 1, since LESO is a linear extended state observer, I would guess ESO is a nonlinear extended state observer. I would like to see a comparison of the two observers.
5. Page 14, Sentence 296, what is AFONTSMC-NDOB?
6. Why was the study not covered to experiments? The authors need to explain the difficulty and limitations of conducting experiments.
This paper is well written, but some statements need to be revised.
Reviewer 3 Report
This paper is a good paper combining theory with simulation. There are only a few small problems to pay attention to.
1. This paper deals with the control problem that occurs when performing manipulation by attaching a 2-degree of freedom manipulator to the drone. The theoretical basis and the simulation results sound good. However, I wonder what purpose the aerial manipulator is actually for in this system.
2. In sentence 296, what does AFONTSMC-NDOB mean? Something is missing.
3. From the stability analysis, it can be seen that the proposed method only ensures the state can be driven into a sliding domain instead of the sliding surface. What causes unreachable sliding surfaces? Please explain it.
4. Pictures need to be normalized. For example, Figure 2 is about to go beyond the layout boundaries. Figure 3 and Figure 4 are not clear.
5. Please double-check the full formula. For example, equation (20) has an extra equal symbol.
good
Reviewer 4 Report
The article is well-written and presented. I think this manuscript is ready for publications.
Minor editing of English language required
Reviewer 5 Report
The idea is nice, the application is highly intriguing, and the manipulator design is well-executed.
The manipulator design is also notably well-defined. Nevertheless, there is large room for improvement in addressing
the motivations of the controller choices more effectively. Besides, results could be better described and improved.
The inlcusion of missing details should be done to make.
I found the presentation of the controllers and contributors to this task to be slightly confusing.
I wonder why the authors discuss the robot hardware when they only provide simulations.
This can be misleading and may cause confusion.
Another concern arises from the assumption that roll and pitch control are not necessary for a quadrotor while hovering.
This assumption should also be reevaluated because the capture of water samples will undoubtedly induce a momentum that the robot will need to compensate for.
How do the authors control the robot when the robot is required to execute alternative trajectories in conjunction with the manipulator? Further details should be provided.
Case 1
Authors stated multiple times: "Other initial conditions are set to zero". Whic conditions are zero? Further details should be provided.
Authors should better clarify why a backstepping integral fast terminal sliding
mode control based on linear extended state observer has to be used for the specific control of the manipulator.
Have they evaluated this controller with basic hardware and demonstrated improved performance with actual motors?
In the simulations, it should be made clear what kind of disturbances the authors consider for their manipulator from a numerical perspective.
How will the authors manage the rapid overshoot they experience when implementing such a control strategy in a real harwdware device?
Case 2
Authors stated multiple times: "All other initial conditions are 0 (which ones?)". Whic conditions are zero? Further details should be provided.
A variance of 0.01 generally suggests a relatively low level of variability within the dataset. Further simulations augmenting such a value should be performed.
In the simulations, it should be made clear what kind of disturbances the authors consider for their manipulator from a numerical perspective.
Case 3
Authors stated "The Gaussian noise is added to simulate the wind gusts". Numerical details should be provided.
Performances: "tracking error between -50 cm and 50 cm".
Tracking errors on the order of a few centimeters or less may be necessary to ensure accurate data collection, especially in this particular application where
precise spots should be checked.
In the simulations, it should be made clear what kind of disturbances the authors consider for their manipulator from a numerical perspective.
Minor Points
1. It seems a repetition: "For our work, in the cable-driven aerial robotic arm in water quality sampling, the hovering state of the
aircraft is treated as a floating platform, only considering the dynamics of the manipulator."
2. "In our work, when the cable-
driven aerial manipulator is in water sampling, the hovering state of the aircraft is treated
as a floating platform, only considering the dynamics of the manipulator. For our work,
in the cable-driven aerial robotic arm in water quality sampling, the hovering state of the
aircraft is treated as a floating platform, only considering the dynamics of the manipulator.
In flight mode, the manipulator serves as the payload of the aircraft, only considering the
dynamics of the aircraft. Therefore, this paper intends to adopt an independent modelling
method to obtain the system model of UAM."
I would suggest rephrasing this sentence to make it simpler to understand.
3. LESO Design should be improved. Symbols are not correct.
Manipulator Controller Design symbols should be revised
4. equation (36) s1 should be derived.
5. Some of the parameters in table 1 should be shown in Figure 5 and better explained in the text, e.g. what are exactly
m2, l1, l2, L?
Minor changes should be done
Round 2
Reviewer 5 Report
Based on the comments, a more extensive numerical campaign would have been expected.
1. Only slight modifications were made to the introduction in order to elucidate the contribution.
2. Any modifications have been made regarding the fact that talking at the beginning about the hardware
could have been misleading
3. As also stated by other reviewers, Remark 2 is too idealistic.
6. Response 6 is quite perplexing to me.
7. Any modification has been made to make clearer the inserted disturbances in the simulations. Besides, based on the comments, a more
extensive numerical campaign would have been expected.
It should be slightly improved
